# Chicken Hypothalamic and Ovarian DNA Methylome Alteration in Response to Forced Molting

**DOI:** 10.3390/ani13061012

**Published:** 2023-03-10

**Authors:** Tongyu Zhang, Chengfeng Li, Jianwen Deng, Yaxiong Jia, Lujiang Qu, Zhonghua Ning

**Affiliations:** 1Department of Animal Genetics and Breeding, College of Animal Science and Technology, China Agricultural University, Beijing 100193, China; 2Hubei Shendan Healthy Food Co., Ltd., Xiaogan 432600, China; 3Institute of Animal Science, Chinese Academy of Agricultural Sciences, Beijing 100091, China; 4National Engineering Laboratory for Animal Breeding, College of Animal Science and Technology, China Agricultural University, Beijing 100193, China; 5State Key Laboratory of Animal Nutrition, Beijing 100193, China

**Keywords:** forced molting, DNA methylation, functional regions, reproductive function, chicken

## Abstract

**Simple Summary:**

Forced molting refers to the systemic stress response of laying hens caused by some externally imposed artificial measures (such as starvation), resulting in production suspension, weightlessness, and molting of laying hens. After the stress is restored to normal feeding conditions, the chickens will recover their physical fitness so that the egg-laying rate can be restored to the second peak of egg production and egg quality can be improved. Previously, we thought that only genes regulate this reversible biological process at the transcriptome level, but further studies showed that environmentally induced epigenetic inheritance could also regulate this process, and there is an interspersed effect between them.

**Abstract:**

Epigenetic modifications play an important role in regulating animal adaptation to external stress. To explore how DNA methylation regulates the expression levels of related genes during forced molting (FM) of laying hens, the hypothalamus and ovary tissues were analyzed at five periods using Whole-Genome Bisulfite Sequencing. The results show that methylation levels fluctuated differently in the exon, intron, 5′UTR, 3′UTR, promoter, and intergenic regions of the genome during FM. In addition, 16 differentially methylated genes (DMGs) regulating cell aging, immunity, and development were identified in the two reversible processes of starvation and redevelopment during FM. Comparing DMGs with differentially expressed genes (DEGs) obtained in the same periods, five hypermethylated DMGs (DSTYK, NKTR, SMOC1, SCAMP3, and ATOH8) that inhibited the expression of DEGs were found. Therefore, DMGs epigenetically modify the DEGs during the FM process of chickens, leading to the rapid closure and restart of their reproductive function and a re-increase in the egg-laying rate. Therefore, this study further confirmed that epigenetic modifications could regulate gene expression during FM and provides theoretical support for the subsequent optimization of FM technology.

## 1. Introduction

The poultry industry is an important pillar of animal husbandry in China. Commercial hens usually begin laying at approximately 100 days and reach the first peak of egg production at approximately 150 days. The egg-laying rate (ELR) can reach more than 90% at this time and then gradually decrease to 60% approximately a year later. Considering the ratio of feed to egg and feeding costs, the breeder will decide whether to eliminate old hens. When the egg market is depressed or laying hens with high ELR is not easy to obtain, forced molting (FM) technology has been applied to improve the economic benefit of overcoming the crisis, which not only improves the utilization period of hens but also improves the ELR [1].

Forced molting refers to chickens producing systemic stress response to some external artificial measures, such as fasting, which causes a halt in egg production, weight loss, and molting. While returning to normal feeding conditions after fasting, the hens recover, leading to the restoration of the ELR to the second peak of egg production (SPEP) and improvement in egg quality [2]. FM also has some shortcomings that directly lead to the failure to implement and restrict the further large-scale promotion of this technology. The highest mortality rate of laying hens during FM can reach 30%, while the success of FM is marked by a mortality rate of less than 0.2%. In addition, the time between the two peaks of ELR is too long, which increases the feeding cost. Moreover, the weight and feed consumption of the layers increase after FM, and the ELR rapidly decreases in the next six months.

The reproductive function of laying hens is diminished and then restarts during FM. Ovary and oviduct cells undergo senescence and apoptosis under stress conditions, causing follicle development to stop, ovarian atrophy [3], and serious feather aging and shedding. The recession of the reproductive organs, such as the ovaries and oviduct, can cause a gradual decline in ELR. After returning to normal feeding, the hypothalamic–ovarian-gonadal axis of laying hens increases the secretion of thyroxine, enhances metabolic function, induces new feather growth, and secretes a large amount of estrogen and progesterone to restore the ELR to the SPEP [4].

Chickens over 500 days old are more likely to produce broken and soft-shelled eggs because of vitamin D3 metabolic disorders, which affect the absorption of blood calcium [5]. After FM, egg and eggshell quality are significantly improved due to the large consumption of body fat and the return of blood calcium metabolism to normal. After the fat deposited in the uterine glands is consumed, eggshell secretion resumes [6]. In addition, during the recovery from molting, the concentration of intestinal calcium-binding proteins increases, promoting the absorption of intestinal calcium and improving eggshell quality, egg weight, and egg quality [7,8]. Eggshell intensity changes can also be caused by cell apoptosis, autophagy, and aging in the oviduct shell gland during fasting, thus causing degeneration of the shell gland tissue. Eventually, this leads to reduced oviduct substances, such as hormones and calcium secretion, causing the shell quality to drop; however, the proliferating and remodeling cells are restored after the feed supply is restored [9,10]. In addition, liver and gut microbes work together to promote nutrient metabolism and absorption, resulting in a healthier body after FM [11].

DNA methylation is a form of epigenetics that plays an important role in cell differentiation, development, aging, and disease by regulating gene expression. DNA methylation has been widely studied in chickens, mainly involving embryonic development [12], meat quality traits [13,14,15], fat deposition [16,17,18], environmental adaptation [19,20], nutritional regulation [21,22], disease resistance [23,24,25], and reproductive traits [26,27,28]. However, these complex biological phenomena also occur during FM. In previous studies, transcriptomic techniques were applied to reveal the dynamics of gene expression during FM [29]. Therefore, this study aimed to explore the dynamic change process of DNA methylation regulating gene expression during FM using whole-genome bisulfite sequencing (WGBS), thus enriching the understanding of the genetic mechanisms of FM reactivation in the chicken reproductive system. 

## 2. Materials and Methods

### 2.1. Individual Selection and Tissue Collection

In this experiment, chickens were taken from the previous study [29] based on the estimated ELR of the chickens at five different periods during FM. Three healthy chickens of the same weight and age were selected as the experimental chickens for each period. The 15 chickens were humanely euthanized. Hypothalamic and ovarian tissues were collected rapidly and snap-frozen in liquid nitrogen. Thus, 30 tissue samples were collected and stored at −80 °C until DNA and RNA extraction.

All animal experiments were performed in accordance with the Guidelines for Experimental Animals established by the Ministry of Science and Technology (Beijing, China) and were approved by the Animal Care Committee of China Agricultural University (Approval ID: XXCB-20090209), Beijing, China. All efforts were made to minimize animal suffering.

### 2.2. Animal Experimental Design

The FM experiment process refers to the previous study [29]. A total of 44,079 Jingfen No. 6 laying hens were selected to perform FM at the laying farm of Hubei Shendan Company (Anlu, China). All chickens were housed in a six-story coop in an enclosed house. The experimental chickens were sampled at five different time points. 

① The laying hens at 224 days of age reached the first peak of egg production, and the ELR was 0.941.

② The ELR decreased to 0.774 when the chickens were 456 days old, and FM was induced by fasting for 12 days with the initial 2 days without water until egg production completely ceased. At the same time, chickens were exposed to 8 h of light and 16 h of darkness (8 L:16 D) during FM. The average room temperature was 18.4 °C, and the humidity was 45.9%.

③ When the chickens were 469 days old, their average percentage of body weight loss was approximately 30% during fasting, and then the fast ended.

④ Feed and water were then gradually supplied to let the layers recover their strength until the ELR returned to 0.472 by 500 days. 

⑤ Finally, when the ELR recovered to 0.873, the SPEP was reached.

Three healthy hens with uniform body weight were selected from the above five different periods and slaughtered. The hypothalamic and ovarian tissue samples were collected for WGBS.

### 2.3. DNA Isolation and WGBS

Genomic DNA was isolated from each of the collected tissues using the QIAamp Fast DNA Tissue Kit (Qiagen, Dusseldorf, Germany) according to the manufacturer’s protocol and quantified using an Agilent 2100 spectrophotometer (Agilent Technologies, Palo Alto, CA, USA). DNA samples were fragmented using sonication and subjected to bisulfite conversion. The Accel-NGS^®^ Methyl-Seq DNA Library Kit (Swift Biosciences, Ann Arbor, MI, USA) was used to attach adapters to the single-stranded DNA fragments. The adapter-ligated DNA was enriched with eight cycles of PCR with the following thermal profile: initial incubation at 95 °C for 2 min; eight cycles of 95 °C for 30 s; 65 °C for 20 s; 72 °C for 45 s and a final incubation at 72 °C for 8 min. The reaction products were purified using the Qiagen Gel Purification Kit (Qiagen, Dusseldorf, Germany) and quantified using the Qubit dsDNA HS Assay Kit (Life Technologies, Carlsbad, CA, USA). An Agilent 2100 spectrophotometer was used to test the integrity of the sequencing libraries. Paired-end 2 × 150 bp sequencing on an Illumina HiSeq 4000 platform (Illumina, San Diego, CA, USA) was then performed at Shanghai Personal Biotechnology Co., Ltd. (Shanghai, China). A 30× sequence depth was achieved for each library using WGBS.

### 2.4. Data Filtering and Identification of Methylated Cytosines

SOAPnuke (version 1.6.0, BGI, Shenzhen, Guangdong, China) and in-house Perl scripts were used to remove reads containing adapter contamination, low-quality bases, or undetermined bases. Sequence quality was then verified using FastQC (http://www.bioinformatics.babraham.ac.uk/projects/fastqc/, accessed on 27 October 2021). Reads that passed quality control were mapped to the chicken reference genome Gallus_gallus-6.0 (https://www.ncbi.nlm.nih.gov/assembly/GCF_000002315.6, accessed on 29 October 2021) using BSMAP (Baylor College of Medicine, Houston, USA, accessed on 3 November 2021). Then, the transformation of C > T and G > A was carried out, and the sequencing results were compared to genomes to select the best pairs to achieve accurate localization of reads. Duplicate reads were removed using the BSMAP software after mapping. The mapping results were then obtained, and the mapping rate and other statistical information were calculated. Each sample of the methylation site was identified using BSMAP, and the methylation sites (methylation levels) were calculated as follows:ML = mC/(mC + umC)
where ML represents the methylation level, mC represents the number of C bases that were methylated, and umC represents the number of C bases that were not methylated.

Then, genome regions were annotated for methylation sites, including the promoter region (2 kb upstream of the transcription start site), 5′UTR, exon, intron, and 3′UTR.

### 2.5. Identification and Distribution of Methylated C Sites

Using the BSMAP software to analyze the methylation site of the sample, the methylation C bases of CG, CHG, and CHH (H stands for A, C, and T) on the genome of the distribution ratio and specific functions regional distribution regularity of methylation level in the process of FM were obtained.

### 2.6. Horizontal Distribution of Methylation in Functional Areas

Different functional regions of the genome play different roles in gene expression. To further count the methylation levels of these functional regions, statistical analysis of site information in each context was performed in various gene functional regions.

### 2.7. Identification of Differentially Methylated Regions (DMRs) and Functional Enrichment Analysis

In this study, R package methylKit (http://www.bioconductor.org/packages/release/bioc/html/methylKit.html, accessed on 7 November 2021) was used to filter the samples and then combined the samples to remove the batch effect. A sliding window approach (10,000 bp window size, 500 bp overlap) was used to screen for genomic regions that were differentially methylated between the different periods in the chicken groups. Only cytosine sites that were covered by at least three reads were included in this analysis. Fisher’s exact test was performed for each window, and *p*-values were corrected for multiple tests using the false discovery rate (FDR). Windows with FDR < 0.05 and a difference of at least two-fold in the methylation level (total mC/total C reads) between the two different chicken groups were identified as DMRs. The genome-wide functional regions of DMRs, including intron, exon, 5′UTR, 3′UTR, promoter, and intergenic regions, were obtained using the R package genomation.

DMGs were annotated on Ensemble (https://asia.ensembl.org/index.html, accessed on 24 November 2021) using chicken as species, based on the physical location in the chromosome of the DMRs. To explore the function of the DMGs, Gene Ontology (GO) and Kyoto Encyclopedia of Genes and Genomes (KEGG) were performed using DAVID (https://david.ncifcrf.gov/, accessed on 29 November 2021). GO terms and KEGG pathways with *p* < 0.05 were considered significantly enriched. Finally, the R package ggplot2 was used to visualize the functions of the different groups of DMGs. 

## 3. Results

### 3.1. Sequencing Data Statistics

To investigate the role of DNA methylation in experimental chickens from five different periods during FM, hypothalamus and ovary tissues from chickens were isolated for WGBS. Thirty samples from five groups (three biological replicates from each group in the hypothalamus and ovary) were used for WGBS. The samples were sequenced to obtain image files, which were converted by the sequencing platform software to generate raw data, namely off-machine data. The conversion efficiency of bisulfite treatment was above 99% when the raw data was filtered to gain clean data. Statistical analyses on raw and clean reads of each sample were performed, including sample names, data size, the percentage of fuzzy bases, Q20 (%), Q30 (%), and mapping rate (Appendix A). The results showed that the clean reads of each sequenced sample were more than 176G. The percentage of fuzzy bases in each sample was less than 1.312 × 10^−3^, Q20 was greater than 93%, Q30 was greater than 87%, and the mapping rates were all greater than 84%. Therefore, the quality of sequencing data was qualified, and subsequent DNA methylation analysis was carried out.

### 3.2. Identification of Methylated C Sites and Their Distribution

In this study, the BSMAP software was used for comparative analysis of methylation sites in each sample. The methylation ratio of cytosine in the genomic DNA of 30 hypothalamic and ovarian tissues ranged from 1.42 to 4.81% after strict screening (FDR < 0.01). In addition, CG-methylated cytosine sites accounted for the highest proportion in all tissue samples, ranging from 76.45 to 94.49%. The proportion of methylated cytosine in the mCHG and mCHH types was the lowest, ranging from 0.94 to 3.68% and 4.61 to 19.86%, respectively (Appendix A). These results indicate that methylation mainly occurred at CG sites during the five different periods of FM.

### 3.3. Cluster Analysis between Samples

In a previous study, the gene expression patterns of the same batch of chickens in groups 1-vs.-2, 2-vs.-3, and 3-vs.-5 during FM were consistent with the change in ELR. Because those three periods (1-vs.-2, 2-vs.-3, and 3-vs.-5) reflect the biological processes that continuously change during FM, the change patterns of DNA methylation in these three groups should be focused on. First, inter-sample cluster analysis was performed on the hypothalamus and ovarian samples, and the results proved that any two groups of samples had more inter-group differences than intra-group differences, which means that the data were suitable for subsequent DNA methylation analysis (Appendix A).

### 3.4. Distribution of Methylation Levels in Functional Regions of the Genome

The methylation levels in different functional regions of the genome also change during FM. CG is the main methyl group in different animals, and DNA methylation plays an important role in maintaining genomic stability and regulating gene expression in the CG environment. Therefore, this study focused on statistical analysis of the average methylation level of CG sites in various genomic functional regions in each context. The focus here was on promoters, 3′UTR, exons, introns, 5′UTR, and repeats. The promoter region was located 2 kb upstream of the transcription start site.

Finally, the distribution of CG methylation levels of functional elements in all samples was obtained (Figure 1). In addition, the variation trend of DNA methylation on functional elements of different genomes was the same in all samples in the mC environment. Among them, intron and 3′UTR DNA methylation levels were the highest, followed by exon and repeat regions, and 5′UTR methylation levels were the lowest. In the promoter region, the proximal region showed the lowest level of DNA methylation, followed by the intermediate region, whereas the distal region showed the highest level of methylation.

### 3.5. DMR Detection

In this study, WGBS was performed on hypothalamus and ovarian tissues at three different periods of FM (1-vs.-2, 2-vs.-3, and 3-vs.-5), and a large number of DMRs were detected, which, to some extent, reflected genome-wide methylation levels. DMRs were then compared to six genomic elements of different functional regions: 5′UTR, 3′UTR, exon, intron, promoter, and intergenic. The average methylation levels in the six different regions were evaluated using hyper-DMR and hypo-DMR (Figure 2). Overall, the changes in the methylation levels in the six functional regions of each group were similar. The majority of DMRs were located in the 5′UTR and 3′UTR regions, and they were also the largest. There were more DMRs in the promoter region but fewer in the intron and intergenic regions. In addition, the number of DMRs in the hyper-methylated region was lower than that in the hypo-methylated region in the hypothalamus from the 3-vs.-5 groups, while the number of DMRs in the hyper-methylated region was higher than that in the hypo-methylated region in the other five groups.

### 3.6. Identification of DMGs and Functional Enrichment Analysis

In this study, WGBS was conducted on the hypothalamus and ovarian tissues in three groups of 1-vs.-2, 2-vs.-3, and 3-vs.-5 to identify DMRs, and gene annotation was conducted to search for DMGs (hypermethylated genes and hypo-methylated genes) through the Ensemble website (Figure 3, Appendix A). The distribution of hyper and hypo DMGs in the 1-vs.-2 group was relatively uniform, while 112 hyper DMGs in the hypothalamus from the 2-vs.-3 group were the largest number among all groups, and 16 hypo DMGs in the ovary from the 2-vs.-3 group were the fewest. Furthermore, 16 DMGs with reversible methylation levels were identified in two reversible processes in the hypothalamus and ovary, and six genes (CD2AP, CERK, FBXW11, YAP1, NUMB, and SLC7A11) were reported to be associated with cellular senescence (Table 1).

The DMGs of the hypothalamus and ovary in these three different periods were annotated for GO and KEGG functions (Figure 4). Considering that hyper DMGs and hypo DMGs often act comprehensively on related pathways, the same groups of hyper and hypo DMGs were mixed for GO and KEGG analyses. The results showed that 123 DMGs in the hypothalamus (1-vs.-2) were mainly annotated on three significant GO terms (*p* < 0.05): positive regulation of lamellipodium morphogenesis, negative regulation of cell size, and positive regulation of transcription from the RNA polymerase II promoter. The 101 DMGs in the ovary (1-vs.-2) were mainly annotated in three significant GO terms (*p* < 0.05): positive regulation of vascular-associated smooth muscle cell migration, sugar:proton symporter activity, and cytosol. The 162 hypothalamic DMGs (2-vs.-3) were annotated in eight significant GO terms: cerebral cortex tangential migration, embryonic morphogenesis, Rho guanyl-nucleotide exchange factor activity, axon guidance, angiogenesis, transcriptional activator activity, RNA polymerase II core promoter proximal region sequence-specific DNA binding, and cytoplasm (*p* < 0.05). The 64 ovarian DMGs (2-vs.-3) were mainly annotated on two significant GO terms, peptide-serine phosphorylation and cell-cell junction (*p* < 0.05). The 136 hypothalamic DMGs (2-vs.-3) were annotated in 10 significant GO terms: principal sensory nucleus of trigeminal nerve development, LIM domain binding, positive regulation of Wnt signaling pathways, filamentous actin, dendritic spine, ruffle, Z disc, protein binding, regulation of actin cytoskeleton, and focal adhesion (*p* < 0.05), while 46 ovarian DMGs (3-vs.-5) were only annotated in positive regulation of filopodium assembly (*p* < 0.05). No significant KEGG pathway was found in any of the above analyses (*p* > 0.05). Thirty-seven hypermethylated and hypomethylated genes related to cell senescence were reported in the hypothalamus and ovary during the three different periods (Appendix A).

### 3.7. The Conjoint Analysis of Differential Expression Genes (DEGs) and DMGs

Hyper DMGs led to a decrease in DEGs expression level and inhibition of the expression of related genes. To further explore how DMGs regulate the expression level of DEGs and affect the FM process in laying hens, a joint analysis of DMGs and DEGs in the hypothalamus and ovary during the same period was performed, and the common candidate genes are shown in the Venn diagram. The results show that there were no common genes in the hypothalamus from the 1-vs.-2 and 2-vs.-3 groups, while two common genes were found in the other four groups (Figure 5). The methylation levels, transcription levels, and gene functions of these eight genes are shown in Table 2. These eight genes regulate cell development, aging, and cancer. Because of their inhibition of gene expression, the present study focused on five candidate genes with hypermethylation: DSTYK (cellular development), NKTR (cellular immunity), SMOC1 (embryonic development), SCAMP3 (cell cancer), and ATOH8 (embryonic development).

## 4. Discussion

In this study, the WGBS sequencing method was used to detect the genome-wide DNA methylation level in hypothalamus and ovarian tissues in five different periods of FM and explored the imprinting rule of FM stress response to the genome. Studies have shown that DNA methylation varies greatly in different base environments. Similar to many mammals, cytosine in the CG environment usually has a high methylation level, which is similar to the results of other poultry studies [24,28,30]. These results suggest that DNA methylation can regulate gene expression and that changes in methylation levels are usually related to the location of the genome. The methylation level of the promoter region in the genome is usually low, and the promoter region is closely related to the initiation and termination of gene transcription. In addition, the methylation levels of CHH and CHG in this study were relatively low, with the proportion of these two methylation types being approximately 10%. We also found that DNA methylation varies greatly among different functional elements of the genome, and the methylation status varies in different functional regions. It has been shown that DNA methylation probably performs its specific gene regulatory function through methylation changes in one or more regions, and different methylation states of the same gene will also have different functional regulatory effects on gene expression [31].

In a previous study, FM achieved the rejuvenation effect in laying hens, and reproductive organs showed senescence under hunger stress and redeveloped after water supply and feeding were restored [29,32]. Further studies have revealed that this process is accompanied by the regulation of DNA methylation. Therefore, the regulation of methylated genes during FM was focused on in this study, especially some genes that regulate cell senescence in these three periods. This study further suggests that many DMGs that regulate cellular aging and anti-aging play key roles in the process of FM. 

Finally, to further explore how DMGs regulate DEGs, a joint analysis of DMGs and DEGs was performed, and five representative functional genes (DSTYK, NKTR, SMOC1, SCAMP3, and ATOH8) were selected. These five genes were hypermethylated to inhibit gene expression and to perform their biological function during three groups (1-vs.-2, 2-vs.-3, and 3-vs.-5). Chickens from the 1-vs.-2 group are a slow aging process, and egg production is reduced due to its defunct ovary. Furthermore, mutation or downregulation of the ovarian DSTYK gene (1-vs.-2) can lead to multiple organ developmental disorders [33]. The aging of the individual is accelerated, and some organs are diseased, so the mortality rate of the weak laying hens is higher. The decrease in NKTR gene expression (2-vs.-3) causes a decline in cellular immune function and induces tumors [34]. In the 3-vs.-5 group, the hunger stress was relieved, and water and fodder were supplied to the laying hens. The organs of the laying hens redeveloped, and some genes that regulate cell development were screened out during this process. Reduced expression of SMOC1 (3-vs.-5) can promote the development of the reproductive tract and ovaries of Muscovy ducks [35]. However, decreased expression of SCAMP3, a regulatory gene in the hypothalamus (3-vs.-5), can inhibit the occurrence of breast and liver cancer, promoting individual health [36]. ATOH8 was also found during the same period, and its reduced expression can promote animal embryo development [37]. Therefore, these five genes can serve as important biomarkers for regulating aging and redevelopment during different periods of FM, as well as key reference genes for subsequent functional verification.

Therefore, DMGs can participate in the expression of DEGs via DNA methylation during FM. Sixteen DMGs were identified in two reversible processes (2-vs.-3 and 3-vs.-5), seven of which regulated the aging and redevelopment of laying hens. Further analysis revealed five hypermethylated DEGs that inhibited gene expression. These genes enable the normal expression or shutdown of genes related to aging and development during FM and promote the rejuvenation of hens.

Of course, the change process of DNA methylation level during forced molting can also be explored from other points of view. For example, the control could be the first time point with the first peak of egg production and then compared with the other four time points to screen out DMGs separately. This analysis process can be used to explore the changes in methylation levels of specific genes in the following four periods. However, DEGs from three groups (1-vs.-2, 2-vs.-3, and 3-vs.-5) had been screened out in a previous study, and these three groups have important biological significance, respectively. Therefore, in order to better explore the relationship between DMGs and DEGs, the analytical process selected for DNA methylation in this study was similar to that of the previous RNA-seq. Of course, it is also hoped that this process can be further studied in the future.

## 5. Conclusions

In conclusion, DNA methylation is involved in the regulation of gene expression during FM. Many DMGs were found in the 1-vs.-2, 2-vs.-3, and 3-vs.-5 groups. Furthermore, 12 DMGs were found in the hypothalamus and 4 in the ovary during two reversible processes (2-vs.-3 and 3-vs.-5). Five more hypermethylated functional genes (DSTYK, NKTR, SMOC1, SCAMP3, and ATOH8) that repress gene expression were identified. Therefore, this study further confirmed that epigenetic modifications play an important role in the process of FM, providing theoretical support for the subsequent optimization of FM technology.

## Figures and Tables

**Figure 1 animals-13-01012-f001:**
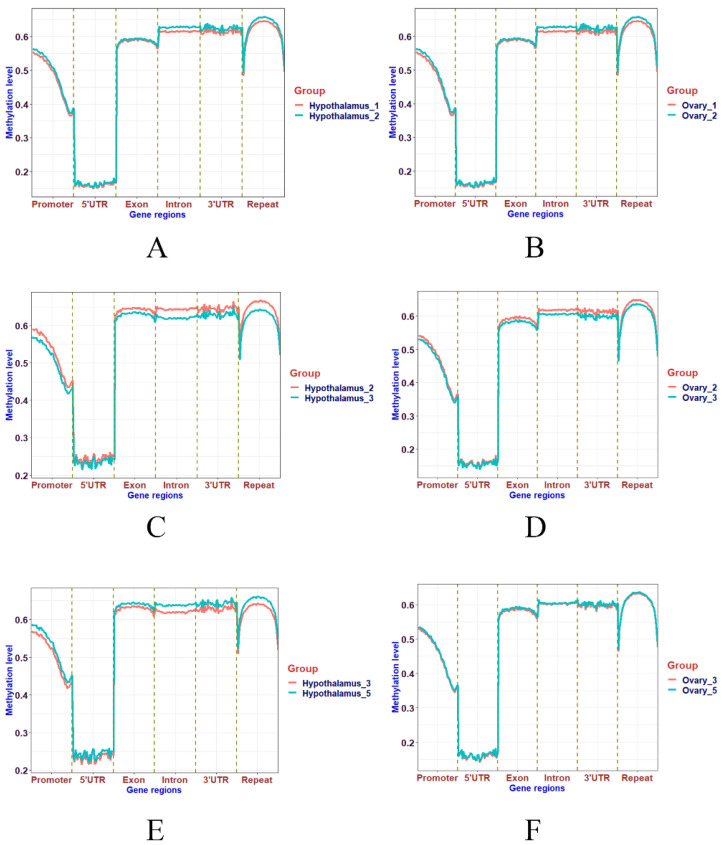
Methylation level at different functional regions in the genome. The horizontal coordinate represents gene functional elements, and the vertical coordinate represents CG-type DNA methylation levels with the hypothalamus (**A**) and ovary (**B**) at group 1-vs.-2, hypothalamus (**C**) and ovary (**D**) at group 2-vs.-3, hypothalamus (**E**) and ovary (**F**) at group 3-vs.-5.

**Figure 2 animals-13-01012-f002:**
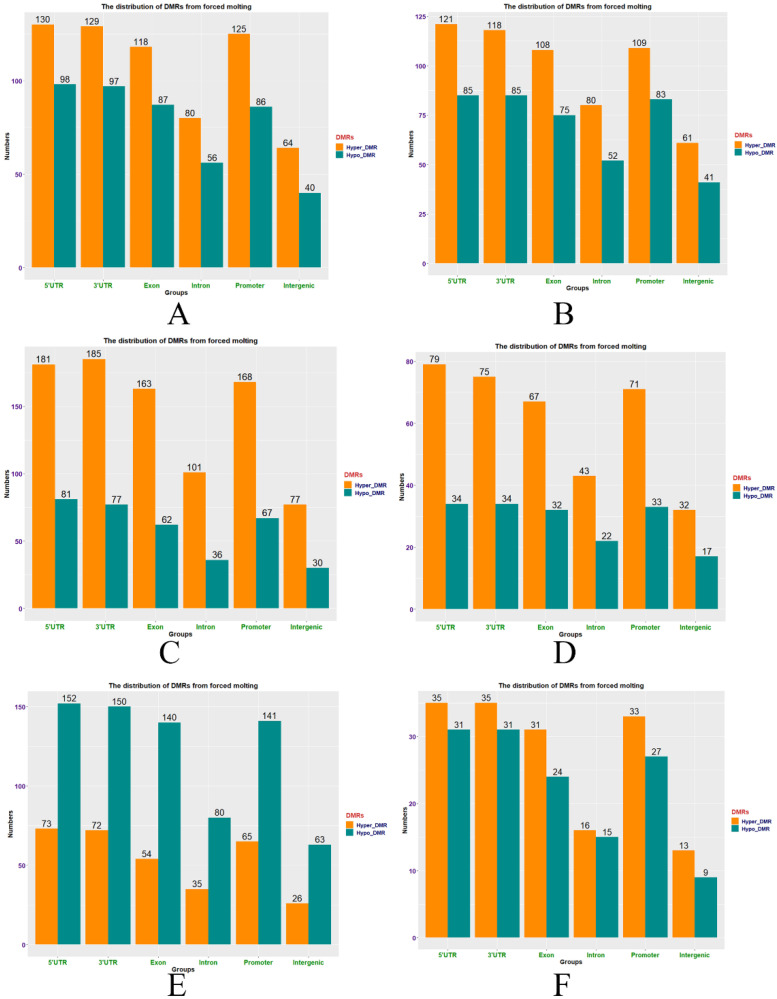
Distribution of DMRs in various genomic elements at different sequence contexts. The horizontal coordinate is the functional region of genes; the vertical coordinate is the corresponding number of DMRs; the colors represent different methylation states. Six figures represent the distribution of DMRs in the hypothalamus (**A**) and ovary (**B**) from 1-vs.-2, hypothalamus (**C**) and ovary (**D**) from 2-vs.-3, hypothalamus (**E**) and ovary (**F**) from 3-vs.-5.

**Figure 3 animals-13-01012-f003:**
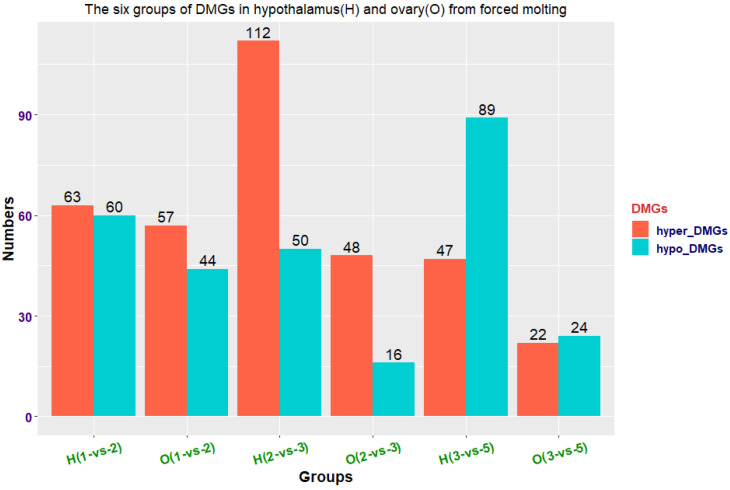
Number of hyper and hypo DMRs in each group and the corresponding number of DMGs.

**Figure 4 animals-13-01012-f004:**
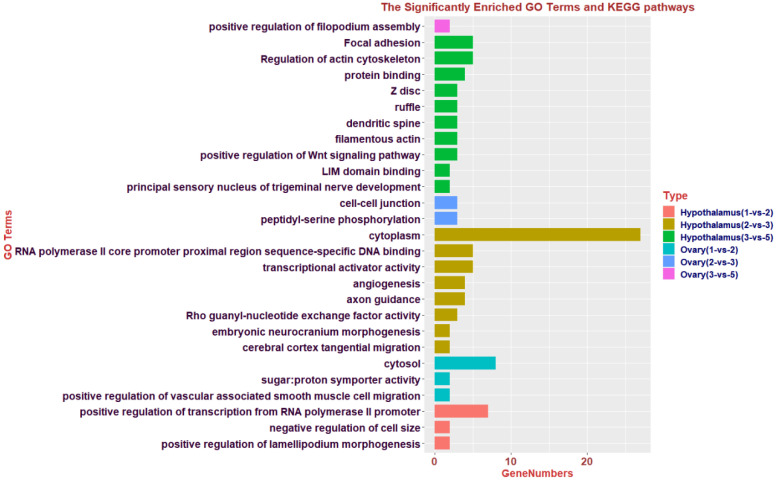
GO terms and KEGG pathways in DMGs from six groups.

**Figure 5 animals-13-01012-f005:**
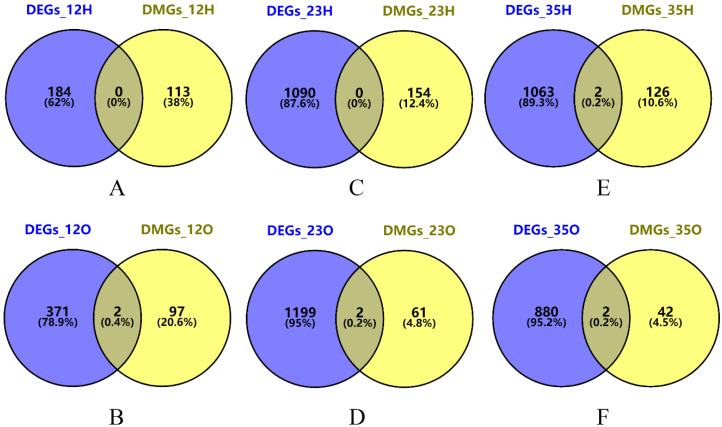
Venn diagram showing six common candidate DEGs and DMGs. DEGs_12H (**A**) and DMGs_12H (**A**) mean the DEGs and DMGs respectively from the hypothalamus in 1-vs.-2 groups; DEGs_12O (**B**) and DMGs_12O (**B**) mean the DEGs and DMGs respectively from the ovary in 1-vs.-2 groups; DEGs_23H (**C**) and DMGs_23H (**C**) mean the DEGs and DMGs respectively from the hypothalamus in 2-vs.-3 groups; DEGs_23O (**D**) and DMGs_23O (**D**) mean the DEGs and DMGs respectively from the ovary in 2-vs.-3 groups; DEGs_35H (**E**) and DMGs_35H (**E**) mean the DEGs and DMGs respectively from the hypothalamus in 3-vs.-5 groups; DEGs_35O (**F**) and DMGs_35O (**F**) mean the DEGs and DMGs respectively from the ovary in 3-vs.-5 groups.

**Table 1 animals-13-01012-t001:** DMGs in two reversible processes during FM in the hypothalamus and ovary.

Tissue	DMGs	Trends in Methylation Level
hypothalamus	CD2AP, CERK, FBXW11, ROS1, UNC13B, YAP1	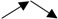
hypothalamus	ABLIM2, ENSGALG00000033770, FSD1, HOXA1, NUMB, SLC7A11	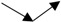
ovary	CD2AP, CPXM1, KCNK13	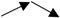
ovary	ENSGALG00000053295	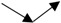

**Table 2 animals-13-01012-t002:** Transcriptional level, methylation state, and function of common DEGs and DMGs in different groups.

Tissues	Groups	Genes	Transcriptional Level	Methylation State	Gene Function
Ovary	1-vs.-2	PRIMA1	−1.518	hypo	proline-rich membrane anchor 1
Ovary	1-vs.-2	DSTYK	−1.001	hyper	dual serine/threonine and tyrosine protein kinase
Ovary	2-vs.-3	NKTR	−1.083	hyper	natural killer cell triggering receptor
Ovary	2-vs.-3	DOCK4	−1.196	hypo	dedicator of cytokinesis 4
Ovary	3-vs.-5	RPS15	−1.680	hypo	ribosomal protein S15
Ovary	3-vs.-5	SMOC1	−1.709	hyper	SPARC-related modular calcium binding 1
Hypothalamus	3-vs.-5	SCAMP3	−1.062	hyper	secretory carrier membrane protein 3
Hypothalamus	3-vs.-5	ATOH8	−1.050	hyper	atonal bHLH transcription factor 8

## Data Availability

All Whole-Genome Bisulfite Sequencing (WGBS) datasets were deposited in the NCBI Short Read database under the accession number PRJNA882134. Refs. [38,39,40,41,42,43,44,45,46,47,48,49,50,51,52,53,54,55,56,57,58,59,60,61,62,63,64,65,66,67,68,69,70,71,72,73,74,75,76,77,78,79,80,81,82,83,84,85,86] are cited in Appendix A.

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
