# Peer review of "Chicken Hypothalamic and Ovarian DNA Methylome Alteration in Response to Forced Molting"

_animals, 2023, doi:10.3390/ani13061012_

Round 1

Reviewer 1 Report

This manuscript describes the ovary and hypothalamus methylome analysis of laying hens subjected to forced molting. Followed previous research of transcriptomes analysis during forced molting, and the manuscript reveal forced molting in a new perspective. The overall research idea is appreciated, and some regulatory DMGs have been found. However, there are some details and contents of the manuscript needs to be improved.

1. (line 44-46) an important indicator of the success of FM technology is a mortality rate of less than 0.2%” please add references. In addition, in hen production, mortality rate could reach 30% during FM, which may make it confusing compared with 0.2%, please explain whether potential factors contributing to increased mortality would affect methylome analysis, and provided mortality data in the current manuscript.

2. (line 54-57) please add references.

3. “After FM, egg and eggshell quality are significantly improved due to the large consumption of body fat and the return of blood calcium metabolism to normal. After the fat deposited in the uterine glands is consumed, eggshell secretion resumes (line 60-62)”. As mentioned above, indicators such as concentration of blood calcium or lipid level et al., should be measured during FM, which may make the research more systematic and comprehensive, so why just collected tissue of hypothalamus and ovaries when design the experiment, and have you measured the related indicators in previous study?

4. Have the formula of feed nutrients changed before and after FM ? Please add the related details.

5. Our experimental (line 85) or “we used ” (line 163), for objectivity of the manuscript,  words such as Our or we are usually not appear.

6. (line 102, line 112) The writing format of ELR were not consistent, with 0.941 and 87.3% respectively, please unify the format. Besides, how could you accurately predict the first peak of egg production (line 101) and second peak of egg production (line 112), and conducted sample selection? second peak of egg productioncould be abbreviated as SPEP.

7. 5-’UTR (line 15) and 3-’UTR (line 15), 5’UTR (line 172) and 3’UTR (line 172). The writing format are inconsistent, please unify the format in the whole manuscript.

8. (Line 206-209) Please explain why you focus on these three periods (1-vs-2, 2-vs-3, and 3-vs-5), and why not 1-vs-3 and 3-vs-5 or other combination, since there exist large difference between point 1 and 3, which may reveal more novel results.

9. A large number of genes related to cellular aging are listed in the table S3, suggesting that the process of forced molting involves processes that regulate cellular aging. In fact, the process of forced molting should also be involved in cell development, so why focus only on the genes responsible for cell aging and not on the genes responsible for cell development? After all, FM is a physiologically reversible process.

10. (Line 424) There is an extra  . here, please delete it.

Author Response

  1. (line 44-46) “an important indicator of the success of FM technology is a mortality rate of less than 0.2%” please add In addition, in hen production, mortality rate could reach 30% during FM, which may make it confusing compared with 0.2%, please explain whether potential factors contributing to increased mortality would affect methylome analysis, and provided mortality data in the current manuscript.

Reply: We have revised this sentence to make it clearer. Our data comes from the production line observation data of enterprises, and there is no references reporting this data at present. If you think the data is unreliable, I can delete this sentence.

  1. (line 54-57) please add references.

Reply: We have added it as requested (line 55).

  1. “After FM, egg and eggshell quality are significantly improved due to the large consumption of body fat and the return of blood calcium metabolism to normal. After the fat deposited in the uterine glands is consumed, eggshell secretion resumes (line 60-62)”. As mentioned above, indicators such as concentration of blood calcium or lipid level et al., should be measured during FM, which may make the research more systematic and comprehensive, so why just collected tissue of hypothalamus and ovaries when design the experiment, and have you measured the related indicators in previous study?

Reply: Our previous study did not measure blood calcium and lipid, but we did measure calcitonin.

Although this is the shortcoming of our experimental design, the variation rule of these indicators has been reported in the literature, so we do not think it is necessary to repeat the measurement.

  1. Have the formula of feed nutrients changed before and after FM ? Please add the related details.

Reply: There are changes before and after the nutritional formula of feed, especially when the feeding is resumed after fasting, relatively rich nutrition is needed to restore the body constitution. However, because the experimental operation time is a little long, we do not have the feed formula of the company at present, and the formula is confidential of the company, so we are very sorry that we cannot provide the feed formula.

  1. “Our experimental” (line 85) or “we used” (line 163), for objectivity of the manuscript, words such as “Our” or “we” are usually not appear.

Reply: We have revised the full text as requested.

  1. (line 102, line 112) The writing format of ELR were not consistent, with 0.941 and 87.3% respectively, please unify the format. Besides, how could you accurately predict the first peak of egg production (line 101) and second peak of egg production (line 112), and conducted sample selection? “second peak of egg production” could be abbreviated as “SPEP”.

Reply: 1) we have unified the format of data.

2) The breeder records this data for the whole coop every day.

3) we have abbreviated “second peak of egg production” as “SPEP” (line 110).

  1. “5-’UTR” (line 15) and “3-’UTR” (line 15), “5’UTR” (line 172) and “3’UTR” (line 172). The writing format are inconsistent, please unify the format in the whole manuscript.

Reply: We have revised the whole text as required.

  1. (Line 206-209) Please explain why you focus on these three periods (1-vs-2, 2-vs-3, and 3-vs-5), and why not 1-vs-3 and 3-vs-5 or other combination, since there exist large difference between point 1 and 3, which may reveal more novel results.

Reply: This is because it corresponds to the previous transcriptome data analysis, and the biological significance of the changes in these three periods is relatively large. More detailed explanations are in line 204-207 and line 372-381.

  1. A large number of genes related to cellular aging are listed in the table S3, suggesting that the process of forced molting involves processes that regulate cellular aging. In fact, the process of forced molting should also be involved in cell development, so why focus only on the genes responsible for cell aging and not on the genes responsible for cell development? After all, FM is a physiologically reversible process.

Reply: Development is a biological norm, but the point of interest in forced molting is that it can rejuvenate chickens, so we are focusing on the genes that regulate the aspects of aging so that we can further analyze the mechanisms of anti-aging.

  1. (Line 424) There is an extra “ . ” here, please delete it.

Reply: we have deleted it (line 401).

Reviewer 2 Report

Line 37 Start sentence with “Forced molting”

Line 42 References are need to support the assertion that FM improves production of sick hens. Delete statement if there are no references

Line 184/185 Thirty samples from 5 groups = 6 samples per group. Three biological replicates per group=15 samples and not 30 samples. Please clarify

Line 324-340 Literature review and can be eliminated. The introduction has enough background information

Line 354-357 References are needed

Line 348-350. Please rephrase sentence. You cannot use “low” twice in the same sentence

Line 358-361. Unrelated to the study. Please delete

Line 366. Change “proves” to “suggests”. The work does not prove this.

Line 372-373. Improve English. A verb is needed

Line 374-375 Delete. Any biological entity will become weak after feed withdrawal

Line 378-379 You cannot establish a causal factor between gene expression and anything. Please modify statement

Line 395 “The current study explored”

Author Response

Line 37 Start sentence with “Forced molting”

Reply: We have revised it as required (line 37).

Line 42 References are need to support the assertion that FM improves production of sick hens. Delete statement if there are no references.

Reply: we had deleted this sentence. Data from the layer breeding observation (line 41).

Line 184/185 Thirty samples from 5 groups = 6 samples per group. Three biological replicates per group=15 samples and not 30 samples. Please clarify

Reply: We have revised it as required (line 182-183).

Line 324-340 Literature review and can be eliminated. The introduction has enough background information

Reply: We have revised it as required (line 322).

Line 354-357 References are needed

Reply: We have revised it as required (line 338).

Line 348-350. Please rephrase sentence. You cannot use “low” twice in the same sentence

Reply: We have revised it as required (line 329-331).

Line 358-361. Unrelated to the study. Please delete

Reply: We have revised it as required (line 338).

Line 366. Change “proves” to “suggests”. The work does not prove this.

Reply: We have revised it as required (line 344).

Line 372-373. Improve English. A verb is needed.

Reply: We have revised it as required (line 350-351).

Line 374-375 Delete. Any biological entity will become weak after feed withdrawal

Reply: We have revised it as required (line 352).

Line 378-379 You cannot establish a causal factor between gene expression and anything. Please modify statement

Reply: We have revised it as required (line 355-358).

Line 395 “The current study explored”

Reply: We have revised it as required to make the expression clearer (line 372-373).
